# Logical–Linguistic Model of Diagnostics of Electric Drives with Sensors Support

**DOI:** 10.3390/s20164429

**Published:** 2020-08-08

**Authors:** Yury Nikitin, Pavol Božek, Jozef Peterka

**Affiliations:** 1Department of Mechatronic Systems, Kalashnikov Izhevsk State Technical University, 426069 Izhevsk, Russia; nikitin@istu.ru; 2Slovak University of Technology in Bratislava, 812 43 Bratislava, Slovakia; pavol.bozek@stuba.sk

**Keywords:** sensors, robotized workplace, algorithm, CNC machine, mechatronic modules, fuzzy inference, diagnostics, optimization

## Abstract

The presented paper scientifically discusses the progressive diagnostics of electrical drives in robots with sensor support. The AI (artificial intelligence) model proposed by the authors contains the technical conditions of fuzzy inference rule descriptions for the identification of a robot drive’s technical condition and a source for the description of linguistic variables. The parameter of drive diagnostics for a robotized workplace that is proposed here is original and composed of the sum of vibration acceleration amplitudes ranging from a frequency of 6.3 Hz to 1250 Hz of a one-third-octave filter. Models of systems for the diagnostics of mechatronic objects in the robotized workplace are developed based on examples of CNC (Computer Numerical Control) machine diagnostics and mechatronic modules based on the fuzzy inference system, concluding with a solved example of the multi-criteria optimization of diagnostic systems. Algorithms for CNC machine diagnostics are implemented and intended only for research into precisely determined procedures for monitoring the lifetime of the mentioned mechatronic systems. Sensors for measuring the diagnostic parameters of CNC machines according to precisely determined measuring chains, together with schemes of hardware diagnostics for mechatronic systems are proposed.

## 1. Introduction

Electric drives based on induction machines are currently standard devices for CNC (Computer Numerical Control) machines and robots. The improvement of the reliability of electrical drives in robots based on induction machines is an important issue which requires resolution for an autonomous system. For problem solving, the diagnostics systems of electrical drives are often used [1,2]. Methods for the diagnostics of induction machines have been discussed in many scientific papers [3,4], as well as progress in the techniques of induction machine diagnosis [5].

Generally, fault diagnosis techniques can be categorized into three main types in accordance with the diagnostic procedures: model-based, signal-based and data-based [6]. Signal processing is an indispensable part of these techniques because the purpose of signal processing is to discover fault signatures from the measured data from machinery in operation [7].

Various IT (information technologies) techniques are increasingly being used for this purpose. Modern information technologies such as data mining [8,9], machine learning [10,11] or neural networks [12] are used very intensively in various social areas and industries. In our paper, we use fuzzy logic techniques from modern IT.

Similarly, the fuzzy method was used by the authors in [13] to diagnose bearings during operation. They stated that, due to the complicated mechanical assemblies, the diagnosis of progressive failures was very difficult. Similarly to our article, these authors introduced the fuzzy rule, which they based on the classification of bearing failures by the clustering method by fuzzy C-means using vibration measurements. In order to detect early bearing disturbances, various statistical features were extracted from this distributed signal of each frequency band. Based on the extracted functions, the fuzzy C-means clustering (FCM) method of fault classification was developed using appropriate member functions, and the basis of fuzzy rules was developed for each tired bearing failure using the marked data.

The diagnosis of rolling bearing failures is considered in [14]. In this paper, a new bearing fault diagnosis method was proposed based on structural feature selection. In contrast to most common fault diagnosis methods, the proposed method transfers the problem of fault diagnosis into a new multi-objective 0–1 programming problem and then uses the MOPSO (multi-objective particle swarm optimization) algorithm to exploit the structural relationship among heterogeneous fault features.

In [15], the authors presented the machine learning technique as a promising tool for the early detection of rolling bearing failure. To solve the problems of false alarms and thus increase the reliability of the detection results, a robust method of early bearing failure detection based on learning of deep transmission was proposed. By training the classifier of supporting vector machines (SVMs), a detection model was created. In the online phase, together with a successive batch of data, the properties of the target deposit were extracted using a common representation obtained in the offline stage and online detection was performed by inserting them into the SVM model. Experimental results showed that the proposed approach improves upon several state-of-the-art detection methods in terms of detection accuracy and false alarm rate.

The authors of the work presented in [16] considered the problem of the left inversion of switched linear systems from the point of view of diagnostics. They used an integrated fuzzy logic system (FLS) that is capable of detecting and isolating abrupt faults which occur in the system.

In [17], a supra-system implementation was proposed to generate an optimized fault diagnostic system. The proposed supra-system is based on the exhaustive comparison of different combinations of fault diagnostic methods and optimized expert systems. It was applied with success to the generation of an expert system for the detection of broken bars, both in a steady-state regime and in a transient state.

The application of artificial intelligence is popular in modern electric drive diagnostics [18,19]; neural networks have even been applied in electrical drive diagnostics [20,21]. Fuzzy logic is currently one of the most popular and progressive techniques for these types of diagnostics and is considered by the authors in [22]. It is necessary to measure diagnostic parameters by sensors in order to perform robot drive diagnostics.

Sensors and sensor systems are used in many industries and have many applications. At present, this does not involve the application of a single sensor but the creation of a system of cooperating sensors. For accuracy with a specific focus, mathematical representations and models for imaging and virtualization are created for this kind of sensory system. The authors in [23] presented a new approach to LiDAR (light detection and ranging) modeling in a virtual test environment, where the sensor was replaced by a sensor model for the development and validation of reliable environmental perception systems for automated driving functions. To determine the properties of materials, the authors in [24] used a shear stress sensor system consisting of distributed embedded piezoelectric polymer films. The shear stress sensing model was mathematically created. This sensor system could visualize the shear strain field and was sensitive to different contact conditions. The authors in [25] described a radar sensor system for automotive vehicles to address the classification of vehicles and pedestrians with an imbalance class with limited experimental radar data available in an automotive radar sensor. The authors in [26] highlighted the fact that each sensor can fail, and therefore it is not recommended to rely on one sensor. The practical solution is to incorporate several competitive and complementary sensors that work synergistically to overcome their shortcomings. The authors in [27] discussed a classification system to evaluate the performance of athletes, with the possibility of a better understanding of the similarity and difference between sports in the case of wheelchair users, where several sensors were used, with one on each wheel axle and one on the frame camber bar.

Another example of the use of sensors concerns applications in buildings. In [28], a methodology was proposed that includes a two-stage approach to improve the use of sensor data for a specific building. When a certain error is reached, the forecasting algorithm (Artificial Neural Network or K Nearest Neighbors) is trained with the most recent data instead of training the algorithm every time. Data collection is provided by a prototype of agent-based sensors.

In [29], the authors stated that inertial motion capture relies on accurate sensor-to-segment calibration. These are cases in which two segments are connected by a hinge joint—for example, in human knee or finger joints as well as in many robotic limbs and thus the joint axis vector must be identified in the intrinsic sensor coordinate systems.

The subject of diagnostics of CNC machines, including robots and their drives, is relevant, and many works have been devoted to its investigation [30,31].

For the purposes of this research, a logical–linguistic model of electrical drive diagnostics was developed. The model is based on fuzzy logic, which deals with the technical condition analysis of an electrical drive. A method for the vibration measurement of an electrical drive was also presented. The integrated parameter, which is defined as the sum of amplitudes of the vibration acceleration of an electrical drive in each of the frequency ranges of a one-third-octave filter, was suggested [32]. The results of the experimental case studies of a robotized CNC workplace are also demonstrated.

The scientific novelty of the research lies in the use of a new diagnostic parameter—the amount of amplitude of vibro-acceleration in the frequency range of 6.3 Hz to 1250 Hz for a one-third-octave filter, which allows the improvement of the accuracy of diagnosis of electric drives of robots and CNC machine tools. In addition, the fuzzy logic rules use a technological criterion—the speed of movement of the robot’s links or machine mechanisms, which also affects the accuracy of the results of the diagnosis. A new criterion for optimizing the continuous diagnostic process was proposed, combining the importance (responsibility) of the node in CNC machine tools or robots and the speed at which degradation processes occur.

## 2. The Logical–Linguistic Model of Electrical Drive Diagnostics

The control system of the electric drive used in this research is a fuzzy proportional integral derivative (PID) and is considered to correspond to fuzzy inference. The mentioned inference is based on the MATLAB Fuzzy Inference System Editor with the Fuzzy Logic Controller toolbox. The control and diagnostic systems can be integrated by using a unified approach to fuzzy logic. The system for diagnostics considers the electrical drive’s features, sensors and modes, and the system for control considers the state of the robot’s drive.

The diagnostics parameters of the robot’s electrical drives acquired by its sensors may include noise, vibration, temperature, electric current and the electromagnetic field. Experiments led to the conclusion that noise cannot be used as a diagnostics parameter because of the amount of noise in the workplace. The temperature of an electrical drive was also not accepted as a diagnostic parameter, because it strongly depends on the temperature of the surroundings within the workshop and is therefore different in various seasons of the year. The electromagnetic field was represented by the sum of the electromagnetic fields of surrounding electrical drives. Thus, the electromagnetic field also cannot be considered for the diagnostics of electrical drives [33].

In the case of the application of frequency inverters within a power electronic subsystem, negative effects that are generated in the subsystem need to be considered. The negative effects are generated as electromagnetic interference, i.e., electromagnetic emissions of low and high frequency.

The monitoring of the technical condition of a robot’s electrical drive is based on a newly developed logical–linguistic model for the diagnostics of electric drives [34]. Equation (1) represents the developed model:(1)Z=F(X, D, T)
where *Z* is the actual technical state of a robot’s drive, *X* is the parameter of integral diagnostics, *D* is the trend of integral diagnostic parameters and *T* is the lifetime of a gear motor.
(2)$$X=\sum x_{i},$$
where *x_i_* represents the vibration acceleration amplitude of an electrical drive including all frequency bands of the one-third-octave filter.
(3)D=G(ΔX, t),
where Δ*X* represents the change of the integral diagnostic parameter over time *t*. The diagnostics model was modeled in the Fuzzy Logic Toolbox package of the MATLAB application. The fuzzy inference system of the electrical drive‘s technical condition evaluation was implemented during the research with three linguistic variables input into the base of the Mamdani-type fuzzy knowledge. The variables are defined as *X*, *D* and *t*. Three terms are applied in every linguistic variable: H—high-value level, M—medium-value level and L—low-value level.

## 3. Results

### 3.1. The Algorithm for the Determination of the Electrical Drives’ Technical State in a Robotized Workplace

***Step 1.*** Set or correct the initial data (matrix R of binary relations between the values of the output parameters B (diagnostic parameters) and the values of the input parameters A (technical condition) [35,36]. The diagram of the impact of defects on diagnostic parameters is shown in Figure 1 [37].

***Step 2.*** Enter the measured diagnostic parameters—vector B.

***Step 3.*** Find vector “A” (presence of errors) to solve the system of equations B = A◦R, where the disjunction is replaced by a maximum and the conjunction is replaced by a minimum.

***Step 4.*** Obtain solution “A” in the form of an interval in which the boundaries define the minimum and maximum solutions.
(4)$$b_{1}=\left(r_{11}\wedge a_{1}\right)\vee \left(r_{12}\wedge a_{2}\right)\vee \left(r_{1m}\wedge a_{m}\right),$$
(5)$$b_{n}=\left(r_{n1}\wedge a_{1}\right)\vee \left(r_{n2}\wedge a_{2}\right)\vee \left(r_{nm}\wedge a_{m}\right)$$

***Step 5.*** Obtain solution “A” in the form of an interval in which the boundaries define the minimum and maximum solutions.

### 3.2. Diagnostics of Mechatronic Modules in a Robotized Workplace

To research and develop a system for diagnosing mechatronic modules (MMs), MM errors are detected, and patterns between errors, modes of operation and diagnostic parameters are analyzed. Based on these laws, the fuzzy logic rules base is established to determine the technical state of the MM.

The allocation of the measured diagnostic parameters ranges in value from −1 to 1. The diagnostic parameters and movement speed fuzzification are consistently performed using the member function of the Gaussian curve. Three terms are defined for every value of the diagnostic parameter and speed, which are equally divided in the range of −1 to 1. The fuzzy rule conclusions are constructed as conditional operators with weights/scales for every rule. The output quantity is determined by the numerical integration, where a technical state level of −1 shows a good technical state (without errors), 0 shows small errors and 1 shows significant errors in MM. To create rules for fuzzy inference, a table is created in which the logical operations between the input parameters are AND operations. Table 1 gives an example of the three diagnostic parameters and speed [38].

Figure 2, Figure 3 and Figure 4 show the examples of the model and simulation results at various speed limits.

In Figure 2a,b and Figure 3a,b, in the vertical axes, a value of −1 corresponds to the minimum value of the diagnostic parameter and a value of +1 corresponds to the maximum value of the diagnostic parameter.

The block diagram of an intelligent MM with a self-diagnostic subsystem is shown in Figure 5.

If the MM state is defect-free, then the monitoring is performed using the fuzzy PID controller. The fuzzification is performed for a fuzzy PID controller by the integral, proportional and differential components of the mismatch error using the fuzzy sets. If the MDM (mechatronic dynamic modul) has been defected, then during the performance of the operation, the degree of developed errors will be considered and a forecast of the possibility of meeting the control objective will be created. Information obtained in our research about defects is transmitted to the operator and to a higher level of control.

### 3.3. Hardware for Diagnosing Robotic Mechatronic Systems

Robotic mechatronic systems must be competitive in terms of quality and cost. This imposes certain limitations on the hardware and software of the diagnostic system. Depending on the organization’s diagnostic method, hardware diagnostic models for mechatronic systems (MSs) are grouped into three groups: parallel, sequential and combined. In a parallel array, the collection and processing of sensor information and the technical state decision of the MS are made in parallel with the computing devices found in each mechatronic module [39]. The computing devices can be microcontrollers or digital signal processors that transmit the solution results to the local MS network.

A CNC or PC can be connected to the LAN (local area network). Sensors are located at the point at which diagnostic signals are generated. Figure 6 describes a scheme of the parallel diagnostic device. With the MS Diagnostic Sequence Organization, the collection and processing of sensor information and technical status decisions are made using a single computing device, which may be a microcontroller, digital signal processor or industrial computer [40,41]. In the research laboratory, the sensors are located at the point at which diagnostic signals are generated. A scheme of the serial diagnostics of the device is shown in Figure 7. A scheme of the combined diagnostics of the device is shown in Figure 8.

The following novel algorithm was designed for the construction of devices for the diagnostics of mechatronic systems, consisting of a sequence of the following steps:Decomposition of mechatronic systems into modules, nodes and elements;Determination of diagnostic parameters in modules, nodes and elements;Selection of sensors to measure diagnostic parameters;Selection of diagnostic intervals.

The algorithm for the construction of MS diagnostic systems is considered based on the example of a researched robotized workplace: a CNC machine. A CNC machine, such as MS, consists of mechanical, electrical, electromechanical and electronic subsystems and CNC equipment; there are also hydraulic and pneumatic subsystems [42].

### 3.4. Decomposition of CNC Machine into Modules, Nodes and Elements

The mechanical subsystem used for the scientific research laboratory consisted of the following components: bearings, caliper, trolley; ball screws; gears, belt drives; spindle units, drive shafts; speed and feed boxes; cooling systems, lubricants; bearings; tool changers; cutting tools; and other parts. The electrical and electromechanical subsystems included the following components: main propulsion engines, power drives, electrical enclosures with electrical equipment and other subsystem elements. The CNC included the following components: drive control systems and feedback sensors.

### 3.5. Definition of Diagnostic Parameters in Modules, Nodes and CNC Machine Elements

Some diagnostic objects are required by CNC lathes and their diagnostic parameters. Table 2 shows some diagnostic parameters given by the research conditions for modules, components and elements of CNC machines [43].

### 3.6. Selection of Sensors to Measure Diagnostic Parameters

When selecting a sensor to measure a diagnostic parameter, it is necessary to take into account the measurement range, the operating conditions of the object during research measurement and availability of measurement techniques. In this case, the measurement range of the diagnostic tools should ensure the registration of the minimum and maximum (limit) values of the diagnostic parameters. The sensor measurement error should be 1–2%. If possible, all sensors—especially vibration and temperature sensors—should be installed in the immediate vicinity of the diagnosed object [44]. The presence of embedded sensors in machine elements and components is ideal, such as position, angular velocity, temperature and vibration sensors; furthermore, a microcontroller is used to convert information to a digital form and perform processing and transfer to other controllers which are built into mechatronic bearings. Table 3 provides diagnostic parameters and the sensors used for their measurement.

### 3.7. Selection of Diagnostic Intervals

The order of analysis of diagnostic parameters depends on the level of responsibility of functional elements of MS, on the time of their diagnosis and on the probability of the appearance of defects therein. For example, in practice, the alignment of the functional element diagnostics sequence occurs in increasing order of the ratio of the time required to diagnose the functional element to the probability of failure of the functional element. The diagnostics interval of functional elements depends on the degree of responsibility of the mechatronic module, the node, the MS element and the speed of the degradation processes. For scientific research and analysis, a general criterion, Ki, is proposed which refers to the level of responsibility of the *i*-th mechatronic module, the node, the MS element and the rate of flow of degradation processes, calculated as
(6)$$K=K_{otv}+K_{degr},$$
where *K_otv_* is the *i*-th element’s responsibility coefficient, ranging from 0 to 0.5 (0.5 is the maximum degree of responsibility), and the *K_degr_* coefficient characterizes the flow rate of degradation processes of the *i*-th functional element, which ranges from 0 to 0.5 (0.5 is the maximum flow rate of degradation processes). A high *K_i_* value means that more critical modules, nodes and elements with a high degree of degradation processes should be diagnosed more frequently. The above factors are determined by the expert estimation method. The approximate values of the coefficients are shown in matrix *K*. The columns in the matrix are arranged in order of increasing rate of the degradation of functional elements (the first column corresponds to a slow degradation rate of the object of diagnosis, while the third column is high). The rows in the matrix are arranged to increase the responsibility of the functional elements:(7)$$K=\begin{array}{ccc} 0.1\ldots 0.3 & 0.4\ldots 0.6 & 0.7\ldots \;1.0\\ 0.2\ldots 0.4 & 0.4\ldots 0.6 & 0.6\ldots 0.8\\ 0.4\ldots 0.6 & 0.6\ldots 0.8 & 0.8\ldots 1.0 \end{array},$$

The diagnostic interval T is calculated according to the formula:(8)$$T=\frac{T_{C}}{K_{i}},$$
where *T_C_* is the time of the diagnostic cycle determined by the hardware and software capabilities of the diagnostic equipment and *K_i_* is a common criterion. Table 4 lists the diagnostic criteria and intervals for modules, components and elements of CNC machines.

Diagnostic intervals are calculated in the last column of Table 4, where *T_min_* is the minimum diagnostic interval. We calculated the sum of diagnostic intervals for modules, components, and elements of CNC machines. According to our original methodology for this research in our example, the sum of diagnostic intervals is 20.9*T_min_*. The multiplicity of the diagnostic intervals is then calculated by dividing the sum by *k* × *T_min_*. The minimum multiplicity is 7.0 and the relative multiplicity is calculated; all values are divided by the minimum multiplicity of 7.0. For practical implementation, the relative multiplicity is rounded to integers. Table 5 shows the frequency of intervals for diagnostics of modules, nodes and elements of CNC machines [45].

For one diagnostic cycle, it is therefore necessary to diagnose, for example, the setting and slides twice, the cutting tool three times and the poppet head once. As an example, we propose a sequence of diagnostic modules, components and elements of CNC machines as follows: 8, 11, 1, 2, 3, 4, 5, 8, 11, 6, 7, 9, 1, 2, 3, 8, 11, 4, 5, 6, 9, 10, and 12.

### 3.8. Multicriterial Optimization of Diagnostic Systems

To solve the optimization of the MS diagnosticsS process problem, the selection of criteria for optimizing the MS diagnostic devices is necessary. This selection is a relatively complex task, given the need to take into account many factors with varying degrees of significance at the same time. A generalized criterion of the optimality of MS diagnostic equipment is defined as the functionality of economic, organizational, technological and technical criteria:(9)$$Y=\left(X_{E},{\;X}_{\mathrm{OT}},{\;X}_{T}\right),$$
where Y is the general optimality criterion of MC, X_E_ is the economic criteria, X_OT_ is the organizational and technical criteria and X_T_ is the technical criteria.

Economic criteria include accident losses, maintenance and repair costs, scrap volume especially in the manufacture of expensive products use of working time (readiness factor), diagnostic costs, etc. It is clear that the above economic criteria should be optimized.

Optimization according to economic criteria is therefore also multi-criteria-based. In the presented research, the economic criterion of optimality—the economic efficiency of using a diagnostic system is defined as the functionality of private economic criteria:(10)$$X_{E}=F\left(X_{1},{\;X}_{2},\;\ldots ,X_{N}\right),$$
where X*_i_* is the subjective economic criteria.

The economic criteria are calculated as the difference in costs of operating the MS without using diagnostic equipment and for operating MS using diagnostic equipment. For example, the economic results of the use of diagnostic systems can be determined by the capital investment efficiency ratio, which expresses the annual savings from the use of diagnostics:(11)$$E=\frac{C_{1}-C_{2}}{\left(K_{1}-K_{2}\right)}$$
where C_1_ and C_2_ are the primary costs of annual production without the diagnosis and with the diagnosis of the MS condition, and K_1_ and K_2_ are the capital expenditures for the production of the annual production of components without the use of the diagnostic system and with diagnostics of the MS condition.

As an example of the selection of organizational and technical criteria, the definition of the time interval is considered for the diagnosis of functional elements depending on the degree of responsibility of the mechatronic module, the node, the MS element and the rate of degradation processes [46,47]. The degradation process refers to depreciation and element destruction, the loss of precision, performance degradation, accumulation of defects, etc. Table 6 shows the feasibility of using diagnostic systems.

A slow rate of degradation of the object of diagnosis is determined by a slow process that causes damage over months or years. The average rate of degradation of an object of diagnosis is determined by processes that cause damage in minutes or hours [48]. During our research, it was found that a high speed or sudden departure of the object of diagnosis is determined by rapid processes that cause damage in seconds or a fraction of a second.

Table 7 shows the feasibility of using diagnostic systems and types of MS maintenance depending on the type of equipment.

The consequences of an accident (destruction) are the main factor determining the feasibility of the use, form and content of the diagnostic system. Another factor is the downtime or availability of spare parts.

## 4. Discussion

The most important prospects for the development of automated technology systems are as follows:Intellectualization;Increased reliability;Modular design.

Through our scientific research methodology, and after many analyses, it was revealed that the diagnostics of technological systems, such as robotized workplaces, increases their level of intellectualization and reliability. The current states of algorithms and software products for automated diagnostic systems reveal a tendency to create diagnostic programs based on modular artificial intelligence methods. The analysis of diagnostic equipment for technology systems has shown that, in the development of small diagnostic devices based on a microcontroller or processor for processing digital signals with excellent computational capabilities and a standard operating system for express-diagnostics that have a connection to an in-depth diagnostics server, the trends in diagnostics are calculated as parameters, the residual lifetime of mechatronic systems and data archiving [49,50].

The research in this field has focused on the spectrum of rolling bearing signals with artificially created damage to the outer ring in the form of transverse grooves. Defects of this type are reflected in the spectrum as peaks in the high-frequency region. When comparing the spectrum of a bearing without a defect and with a defect, it can be observed that the spectrum of a bearing with a defect increases the total vibration level by 0.7–0.8 V, while there are wide peaks at 250 Hz, which correspond to the rolling frequency of rolling elements on the outside ring, and 340 Hz, which correspond to the rolling frequency of the rolling elements on the inner ring.

The presence of errors increases the amplitude by an average of 0.75 V across the frequency spectrum. The vibration level increases by 1.5–2 V at the rotational frequencies of the rolling elements on both the outer ring and the inner ring, while under radial loading, the amplitude increases at the rotor speed. The generated vibration peaks appear even at the frequency of a 50 Hz rotation of the separator.

In our research, neural network modeling was performed in the MATLAB software product (by MathWorks, Natick, USA). The input data in all examples are presented in the form of a two-dimensional vector, including the frequency and the corresponding amplitude:The rotational frequencies of the monitored mechatronic system,The frequency of the noise of the balls on the outer ring,The noise frequency on the inner ring,The frequency of rotation of the bearing rolling elements.

In this work, we used a new diagnostic parameter, which was calculated as the sum of the amplitude of the vibrating range from a frequency of 6.3 Hz to 1250 Hz for a one-third-octave filter. As a result, the accuracy of diagnosing the robot’s electric drives and machines was improved. Based on the rules of fuzzy logic, a technological criterion was used, such as the speed of movement of the robot’s links or machine mechanisms, which also led to an increase in the accuracy of the diagnostic results. The continuous process of diagnosing a machine or robot was optimized on the basis of a criterion combining the responsibility of the units and the speed at which the degradation processes occurred.

Continuous diagnostic systems are recommended for diagnosing the critical propulsion of technological systems, accidents that can lead to human casualties, technological disasters or significant economic damage. It should be noted that diagnosis is conditional on the use of types of sensors that are capable of measuring both electrical and non-electrical quantities.

The article also presents electric drives based on induction machine one of the types of electric motors. During the development of a logical–linguistic model for the diagnostics of electric drivers with sensor support, it was also confirmed that, thanks to its good performance and the development of microprocessor control, this method will quickly gain popularity. The article discusses the strategy of using sensors for tracking diagnostics with the aim of phasifying torque irregularities in the electrical drives of CNC machines and robots. The sensor system works based on the position of the robot rotor or CNC machine. Thus, engine failures and diagnostic parameters are analyzed from the point of view of the programmed technological process. A mathematic model for electric engine diagnostics and its implementation in MATLAB software based on fuzzy logic is presented.

## 5. Conclusions

The following conclusions can be drawn from our work:(1)A new diagnostic parameter has been used, which was calculated as the sum of the amplitude of a vibrating range from a frequency of 6.3 Hz to 1250 Hz for a one-third-octave filter;(2)Technological criteria for fuzzy logic rules were used, such as the speed of movement of the robot’s links or machine mechanisms;(3)Both new approaches have led to improved accuracy regarding the results of the diagnosis of electric drives for robots and CNC machines;(4)The continuous process of diagnosing a machine or robot was optimized on the basis of a criterion combining the responsibility of the units and the speed at which the degradation processes occur.

In future, it will be necessary to expand our research with practical measurements oriented on the multi-criterial optimization of diagnostic systems.

## Figures and Tables

**Figure 1 sensors-20-04429-f001:**
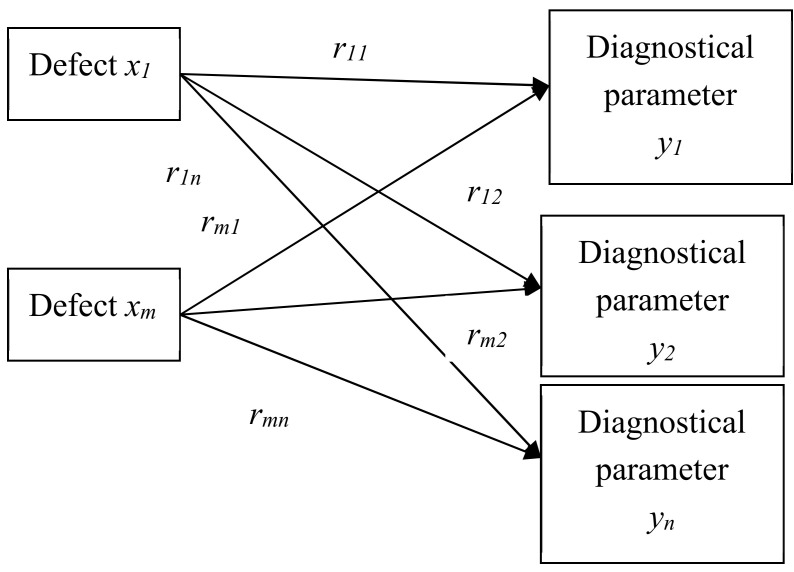
Diagram of the impact of defects on diagnostic parameters.

**Figure 2 sensors-20-04429-f002:**
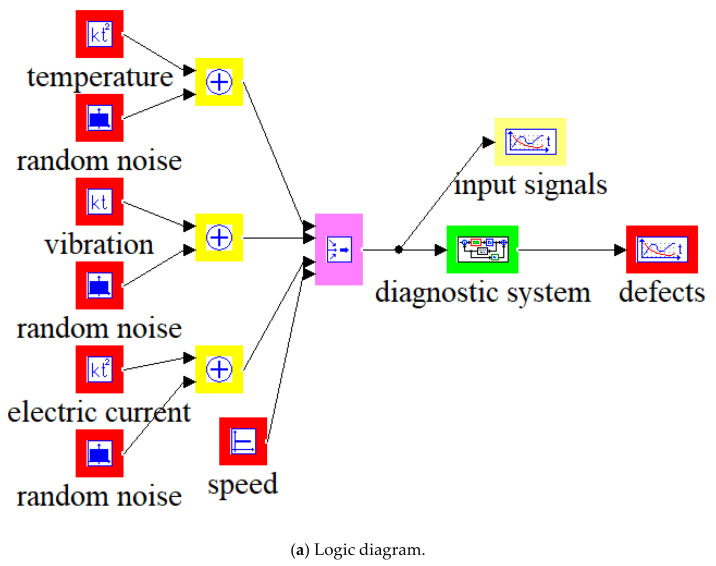
Model of fuzzy inference system for diagnostics and simulation results.

**Figure 3 sensors-20-04429-f003:**
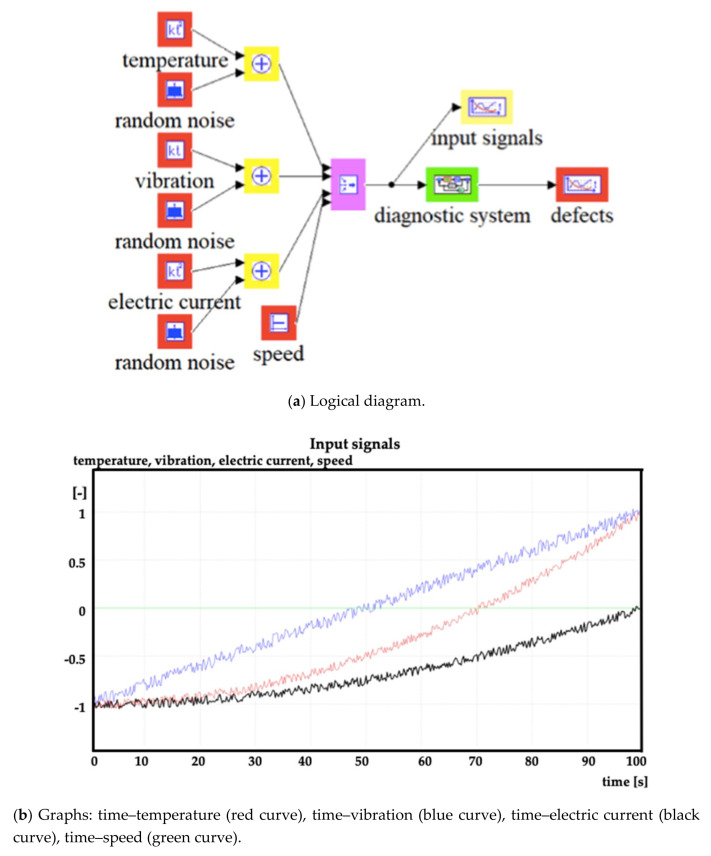
Model of the system for diagnosing fuzzy inference and results of simulation at medium speed.

**Figure 4 sensors-20-04429-f004:**
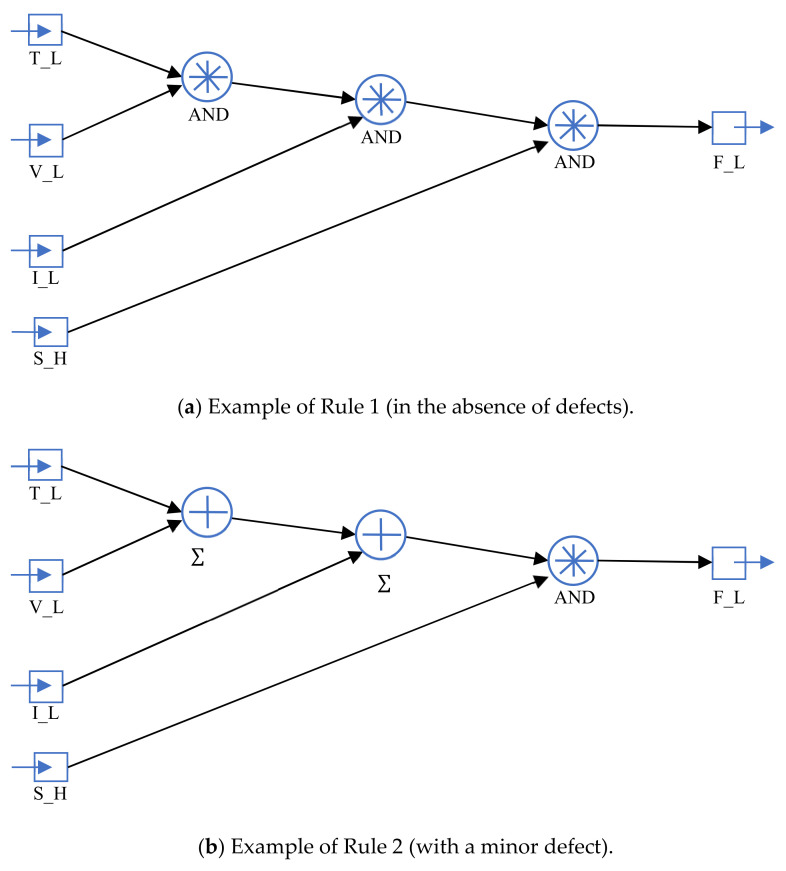
Fuzzy derivation of model rules in the absence of defects and with minor defects.

**Figure 5 sensors-20-04429-f005:**
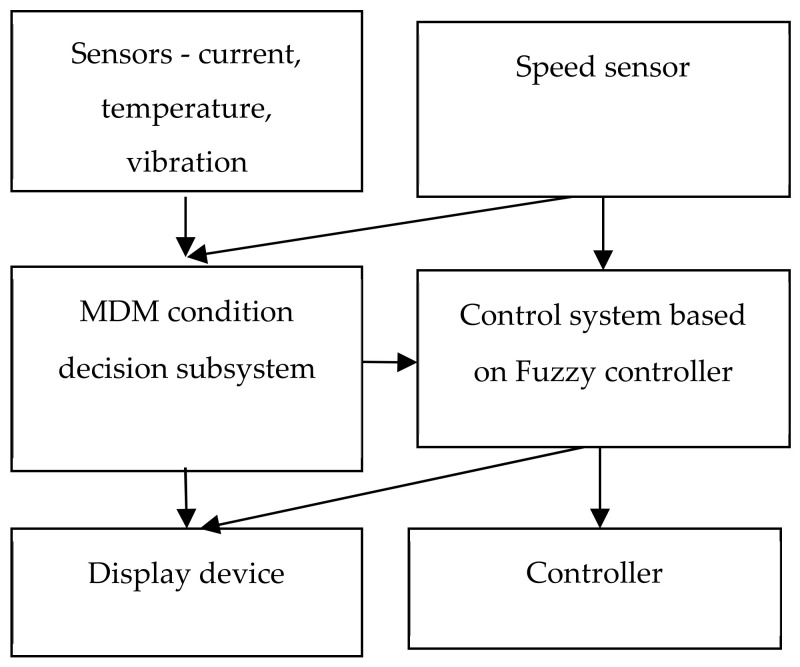
The block diagram of an intelligent MM with a self-diagnostic subsystem.

**Figure 6 sensors-20-04429-f006:**
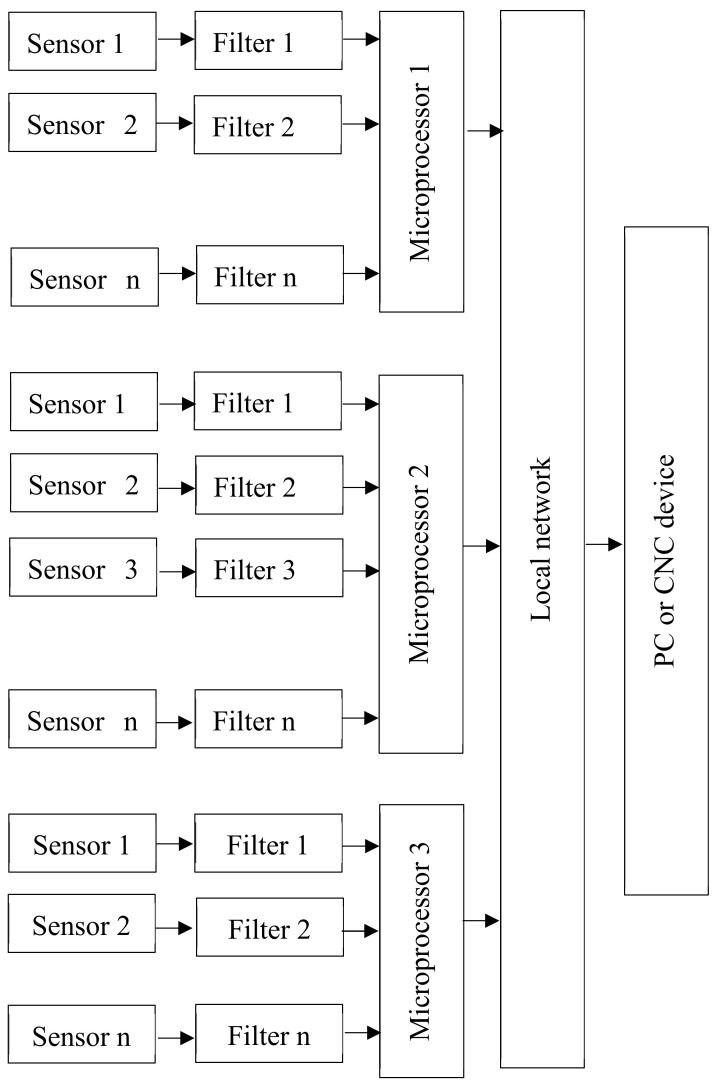
Parallel diagnosis device scheme for PC or CNC: Computer Numerical Control.

**Figure 7 sensors-20-04429-f007:**
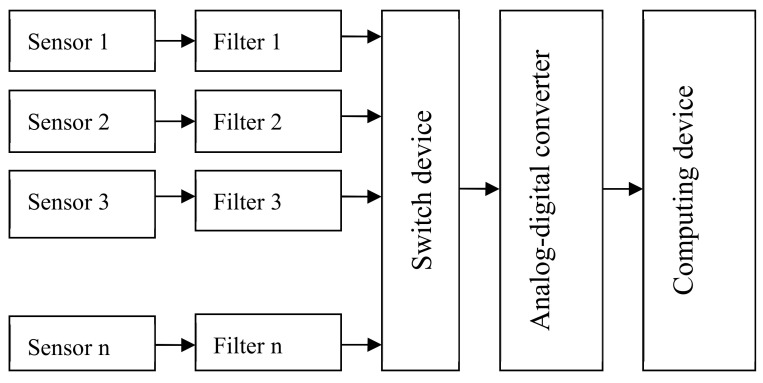
Serial diagnosis device scheme.

**Figure 8 sensors-20-04429-f008:**
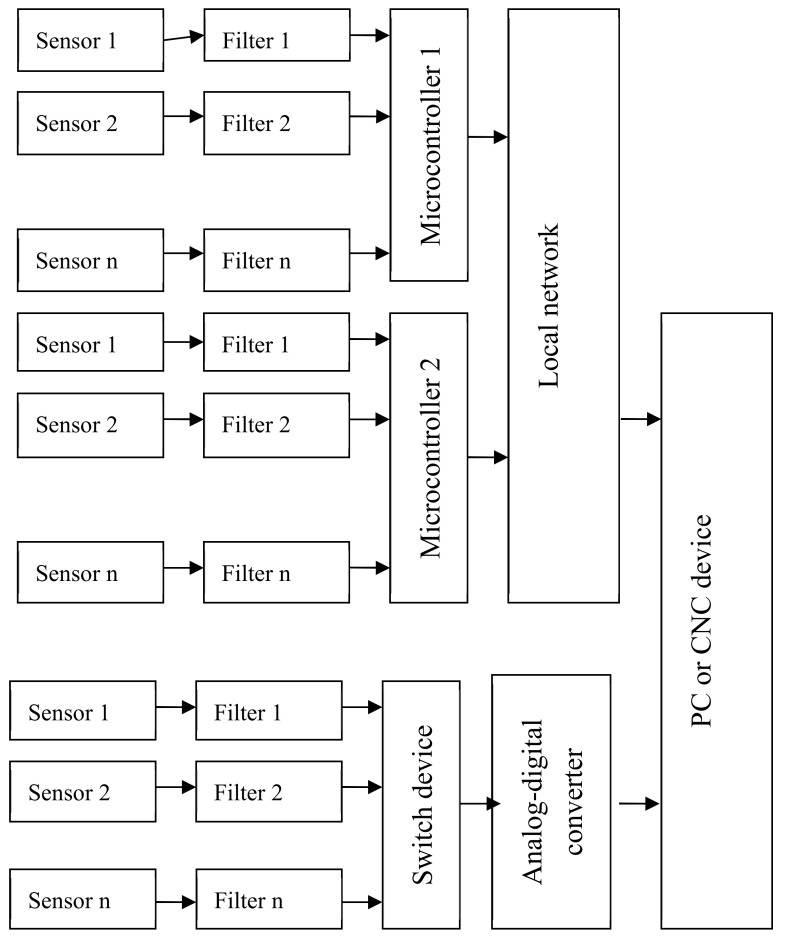
Combined diagnosis device scheme.

**Table 1 sensors-20-04429-t001:** Defect dependence on MDM (mechatronic dynamic modul) parameters and speed.

No Conditions	Temperature	Vibration	Current	Speed	Defect Appearance
1.	L	L	L	H	L
2.	M	M	M	M	M
3.	H	H	H	L	H

Terms for the current technical state of mechatronic modules (MMs): L—defect-free; M—with small defects; H—with significant defects.

**Table 2 sensors-20-04429-t002:** Diagnostic parameters for modules, components and elements of CNC machines.

No	Module, Node, CNC Machine Element	Diagnostical Parameters
1.	Supporting frame of tools	Temperature, motion parameters, power parameters, time intervals, spatial position accuracy
2.	Helical gears	Temperature, motion parameters, power parameters
3.	Gears	Vibration, dynamic parameters
4.	Belt drives	Vibration, dynamic parameters
5.	Spindle units	Temperature, vibration, motion parameters, spatial position accuracy
6.	Bearings	Temperature, vibration, accuracy of spatial positions
7.	Tool holder or tool changer	Temperature, vibration, motion parameters, spatial position accuracy
8.	Cutting tools	Temperature, vibration, accuracy of spatial positions, power parameters
9.	Electromotor	Current, voltage, power, temperature, vibration, movement parameters
10.	Drive control systems	Current, voltage, power, temperature
11.	Sensors	Motion parameters, time intervals
12.	Poppet head	Temperature, spatial accuracy

**Table 3 sensors-20-04429-t003:** Diagnostic parameters for modules, components and elements of CNC machines and sensors for their measurement.

No	Diagnostical Parameters	Sensors
1.	Electrical current	Current sensors up to 50 A, operating frequency 0–25 kHz, Sensor range 0–50 A
2.	Electrical voltage	Voltage sensors 10–500 V, operating frequency 0–25 kHz, Sensor range 0–500 V
3.	Power	Power sensors 0.5–20 kW, operating frequency 0–25 kHz, Dynamic range 90 dB
4.	Temperature	Temperature sensors 0–150 °C
5.	Motion parameters	Accelerometers ± 2 g, encoders 10,000 pulses/rotation, Sensor range 0–10 m/s
6.	Performance parameters	Tensile force sensors up to 10 Kn, Sensor range 0–10 Kn
7.	Time intervals	Timers in the controller, Sensor range 0.1 ms–1 s
8.	Vibration	Accelerometers ± 2 g, operating frequency 1–25 kHz, Sensor range 0–2 g
9.	Spatial position accuracy	Encoders 10,000 pulses/rotation, Sensor range 1–10,000 pulses/rotation

**Table 4 sensors-20-04429-t004:** Criteria and intervals for diagnostics of modules, nodes and elements of CNC machines.

No	Module, Node, Element	*K_otv_*	*K_degr_*	*K_i_*	*T_i_*	*k* × *T_min_*
1.	Setting and wiring	0.5	0.1	0.6	1.67	1.5*T_min_*
2.	Ball helix	0.4	0.2	0.6	1.67	1.5*T_min_*
3.	Cog-wheel	0.2	0.3	0.5	2.00	1.8*T_min_*
4.	Belt gears	0.2	0.3	0.5	2.00	1.8*T_min_*
5.	Spindle units	0.4	0.3	0.7	1.43	1.3*T_min_*
6.	Bearing	0.3	0.3	0.6	1.67	1.5*T_min_*
7.	Tool holder or tool changer	0.2	0.2	0.4	2.50	2.3*T_min_*
8.	Cutting tool	0.4	0.5	0.9	1.11	1.00*T_min_*
9.	Electric motors	0.2	0.3	0.5	2.00	1.8*T_min_*
10	Drive control systems	0.2	0.2	0.4	2.50	2.3*T_min_*
11	Sensors	0.5	0.3	0.8	1.25	1.1*T_min_*
12.	Poppet head	0.1	0.2	0.3	3.33	3.0*T_min_*

**Table 5 sensors-20-04429-t005:** Frequency of intervals for diagnostics of modules, nodes and elements of CNC machines.

No	Module, Node, Element	Failure Rate	Relative Failure Rate	Average Failure Rate
1.	Setting, slide	13.9	2.0	2
2.	Ball helix	13.9	2.0	2
3.	Cog-wheel	11.6	1.7	2
4.	Belt gears	11.6	1.7	2
5.	Spindle units	16.1	2.3	2
6.	Bearing	13.9	2.0	2
7.	Tool holder or tool changer	9.1	1.3	1
8.	Cutting tool	20.9	3.0	3
9.	Electric motors	11.6	1.7	2
10.	Drive control systems	9.1	1.3	1
11.	Sensors	19.0	2.7	3
12.	Poppet head	7.0	1.0	1

**Table 6 sensors-20-04429-t006:** Possibility of using diagnostic systems.

Accident Cost (Destruction)	Slow Rate of Degradation Object of Diagnosis	Average Rate of Degradation Object of Diagnosis	High Degree of Degradation or Sudden Departure of the Subject of Diagnosis
Significant accident costs (destruction)	Portable diagnostic devices *K_otv_* = 0.1, *K_degr_* = 0.1	Portable diagnostic devices *K_otv_* = 0.1, *K_degr_* = 0.5	Stationary diagnostic devices *K_otv_* = 0.1, *K_degr_* = 0.9
Average cost of accident consequences (destruction)	Portable diagnostic devices *K_otv_* = 0.5, *K_degr_* = 0.1	Stationary diagnostic systems *K_otv_* = 0.5, *K_degr_* = 0.5	Continuous protection and diagnostics systems *K_otv_* = 0.5, *K_degr_* = 0.9
High accident costs (destruction)	Stationary diagnostic systems *K_otv_* = 0.9, *K_degr_* = 0.1	Continuous protection and diagnostics systems *K_otv_* = 0.9, *K_degr_* = 0.5	Continuous protection and diagnostics systems *K_otv_* = 0.9, *K_degr_* = 0.9

**Table 7 sensors-20-04429-t007:** Possibility of using diagnostic systems and types of mechatronic system (MS) maintenance.

Type of Device	Possibility of using MS Diagnostic Systems	Types of MS Service	Type of Device
Auxiliary, duplicated, periodically used equipment	Diagnostic and periodic diagnostics are not required for portable diagnostic devices	Repairs	Auxiliary, duplicated, periodically used equipment
Relevant main equipment	Portable diagnostic equipment, stationary diagnostic systems	Service according to the technical condition	Relevant main equipment
One highly responsible device	System of continuous protection and diagnostics	Maintenance, continuous protection	One highly responsible device

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
