# Peer review of "Logical–Linguistic Model of Diagnostics of Electric Drives with Sensors Support"

_sensors, 2020, doi:10.3390/s20164429_

Round 1

Reviewer 1 Report

The authors present a system to diagnose electric drives. The first significant problem of work is its scientific impact. A contribution is neither clearly presented nor confronted to state of the art. The domain survey in the introduction is extensive, but most of the references do not correspond to the core of the article. In the following part, the well-known knowledge is mixed up with the authors' input. The implementation details are mixed with a general idea. The work has a strange structure that, together with the above problems, makes it very hard to take from it. I ask to rewrite the work completely taking into account the above remarks.

Additionally, please take care of the quality of the figures. Some of them have spelling errors (figure 7 - convertor), other a lot of free space (figure 4), some are unclear due to rasterization.

In general, the paper is of poor quality, but the main problem is not the editorial details, but the general scientific considerations. I mentioned that in my review:

The first significant problem of work is its scientific impact. A contribution is neither clearly presented nor confronted to state of the art. The domain survey in the introduction is extensive, but most of the references do not correspond to the core of the article. In the following part, the well-known knowledge is mixed up with the authors' input. The implementation details are mixed with a general idea. The work has a strange structure that, together with the above problems, makes it very hard to take from it. I ask to rewrite the work completely taking into account the above remarks.

In my opinion, at first, the authors should justify the scientific impact in the way I mentioned in the review then we can concentrate on other aspects.

Author Response

Response to Reviewer 1 Comments

Point 1: The authors present a system to diagnose electric drives. The first significant problem of work is its scientific impact. A contribution is neither clearly presented nor confronted to state of the art. The domain survey in the introduction is extensive, but most of the references do not correspond to the core of the article. In the following part, the well-known knowledge is mixed up with the authors' input. The implementation details are mixed with a general idea. The work has a strange structure that, together with the above problems, makes it very hard to take from it. I ask to rewrite the work completely taking into account the above remarks.

Response 1: The work has been rewritten taking into account the above remarks. Words and phrases were added throughout the text emphasizing one's own research and scientific orientation. Further, a scientific contribution has been added, Line 115 – 122: “The scientific novelty of the research lies in the  use of a new diagnostic parameter - the amount of amplitude of vibro-acceleration in the range of 6.3 Hz frequency to 1250 Hz frequency for 1/3-octave filter, which allows to improve the accuracy of diagnosis of electric drives of the robots and CNC tools. In addition, the fuzzy logic rules use a technological criterion - the speed of movement of the robot's links or machine mechanisms, which also affects the accuracy of the results of diagnosis. A new criterion for optimizing the continuous diagnostic process was proposed, combining the importance (responsibility) of the node in a CNC tools or robots and the speed at which degradation processes proceed.”.

Point 2: Additionally, please take care of the quality of the figures. Some of them have spelling errors (figure 7 - convertor), other a lot of free space (figure 4), some are unclear due to rasterization.

Response 2: Thanks for note. We redesigned figure 4, figure 7 and figure 8. So, the word “convertor” we have replace to “Switch device”.

Point 3: In general, the paper is of poor quality, but the main problem is not the editorial details, but the general scientific considerations. I mentioned that in my review:

Response 3: We have fulfilled our scientific considerations, Line 374-381: “The studies used a new diagnostic parameter, which was calculated as the sum of the amplitude of vibrating range from a frequency of 6.3 Hz to a frequency of 1250 Hz for a 1/3-octave filter. As a result, the accuracy of diagnosing the robot's electric drives and machines has been improved. In the base of the rules of fuzzy logic, a technological criterion was used, such as the speed of movement of the robot's links or machine mechanisms, which also led to an increase in the accuracy of the diagnostic results. The continuous process of diagnosing a machine or robot was optimized on the basis of a criterion combining the responsibility of the units and the speed at which the degradation processes proceed.”

Point 4: The first significant problem of work is its scientific impact. A contribution is neither clearly presented nor confronted to state of the art. The domain survey in the introduction is extensive, but most of the references do not correspond to the core of the article. In the following part, the well-known knowledge is mixed up with the authors' input. The implementation details are mixed with a general idea. The work has a strange structure that, together with the above problems, makes it very hard to take from it. I ask to rewrite the work completely taking into account the above remarks.

Response 4: We agree, and we also perceive, that scientific articles may have a broader environment and, of course, the very core of the problem. The surroundings serve to anchor the problem in a given area. Its breadth is a matter of opinion of both authors and opponents. It can be agreed that some articles are beyond the core of the problem. Links outside the core of the issue, in our sole discretion, have been removed (numbers before): [5],[7],[33].

[5] Černecký, J.; Valentová, K.; Pivarčiová, E.; Božek P. Ionization impact on the air cleaning efficiency in the interior. Measurement Science Review, 2015, 15, 4, pp. 156-166.

[7] Qazizada, E.; Pivarčiová, E. Reliability of parallel and serial centrifugal pumps for dewatering in mining process. Acta Montanistica Slovaca. 2018, 23, 2, pp. 141-152.

[33] Baranov, M.; Bozek, P.; Prajová, V.; Ivanova, T.; Novokshonov, D.; Korshunov, A. Constructing and calculating of multistage sucker rod string according to reduced stress. Acta Montan. Slovaca. 2017, 22, 2, pp: 107-115.

We have also re-evaluated the literature review and added references (the current numbering): [6,7], [13-17].

Of course, the literature has been renumbered throughout the text of the article.

Point 5: In my opinion, at first, the authors should justify the scientific impact in the way I mentioned in the review then we can concentrate on other aspects.

Response 5: We emphasized the scientific impact in the form of bullets and placed it in Chapter 5. Conclusions:

- A new diagnostic parameter has been used, which was calculated as the sum of the amplitude of vibrating range from a frequency of 6.3 Hz to a frequency of 1250 Hz for a 1/3-octave filter.

- Technological criteria for fuzzy logic rules were used, such as the speed of movement of the robot's links or machine mechanisms.

- Both new approaches have led to improved accuracy of the results of diagnosis of electric drives for robots and CNC machines.

- The continuous process of diagnosing a machine or robot was optimized on the basis of a criterion combining the responsibility of the units and the speed at which the degradation processes proceed.

Reviewer 2 Report

Line 15: "is composed from the sum" - composed of ...

Line 39: "At present, it is not a question of applying a single sensor and creating a system of cooperating sensors." 

"it is not a question..." but ...

Line 96: "t is the lifetime of a gear motor." should be "T" generally 2nd part of 2nd paragraph needs to be checked about T and t as time or vector from equation 1. It might be a good solution to change vector T for different lette so it would not be so confusing.

figure 2. - interference model shoulod be a graph/scheme not a print screen. same fith consecutive figures. This way it would be smaller but more transparent. Also you show thansients of speed, current, vibration and speed. Which one is which ? there is no description. And what are the units ?

figure 4. definitelly a graph, not a print screen

Table 2. "Worm gears" - worn ? warm ? Worm is an animal.

Table 3. sensors range.

Are you sure you have such huge motor that you need so "big" sensors ? Remember that if you measure relativelly small value at high range, you multiply the overall error.

Assuming you have power sensor range of 20kW, if tou have voltage of 500V to obtain 20kW you need 40A current. That is 40% of measurement range of current sensor.

Author Response

Response to Reviewer 2 Comments

Point 1: Line 15: "is composed from the sum" - composed of ...

Response 1: Thanks for note. We agree, “from” has been replaced on “of”.

Point 2: Line 39: "At present, it is not a question of applying a single sensor and creating a system of cooperating sensors."

"it is not a question..." but ...

Response 2: Now, it is Line 43. Of course, thanks for note. We agree “and” has been replaced on “but”.

Point 3: Line 96: "t is the lifetime of a gear motor." should be "T" generally 2nd part of 2nd paragraph needs to be checked about T and t as time or vector from equation 1. It might be a good solution to change vector T for different lette so it would not be so confusing.

Response 3: Now, it is Line 107. Yes, we agree, “t” has been replaced on “T”. We checked the letters “t” and “T” in the text and in the equations.

Point 4: figure 2. - interference model shoulod be a graph/scheme not a print screen. same fith consecutive figures. This way it would be smaller but more transparent. Also you show thansients of speed, current, vibration and speed. Which one is which ? there is no description. And what are the units ?

Response 4: Thanks for the comment. We have added a description of the figure 2 (and Figure 3 too) and we divided the pictures into three parts a), b) and c). Th colored curves have been explained.

We have entered the units: The horizontal axes show the seconds [s]. On the vertical axes, the units of measurement are given in relative units. A value of -1 corresponds to the minimum value of the diagnostic parameter. A value of +1 corresponds to the maximum value of the diagnostic parameter.

Point 5: figure 4. definitelly a graph, not a print screen

Response 5: Figure 4 was converted to a graph.

Point 6: Table 2. "Worm gears" - worn ? warm ? Worm is an animal.

Response 6: Thanks for the note. We fixed it right “Helical gears”.

Point 7: Table 3. sensors range.

Response 7: We have added their range values for individual sensors.

Point 8: Are you sure you have such huge motor that you need so "big" sensors ? Remember that if you measure relativelly small value at high range, you multiply the overall error.

Assuming you have power sensor range of 20kW, if tou have voltage of 500V to obtain 20kW you need 40A current. That is 40% of measurement range of current sensor.

Response 8: Yes, we agree that the full range of sensors should be used. We have changed the measuring range of the current sensor to 50 A.

Round 2

Reviewer 1 Report

Authors improved the paper by applying the reviewers' remarks. The main problem of its current form is figures quality. Figure 2,3,6 are blurry. Additionally, I recommend extensive English check, e.g.,
in the first sentence of abstract:
"The presented paper scientific discusses"

Author Response

Response to Reviewer 1 Comments -ROUND 2

Comments: Authors improved the paper by applying the reviewers' remarks. The main problem of its current form is figures quality. Figure 2,3,6 are blurry. Additionally, I recommend extensive English check, e.g.,in the first sentence of abstract: "The presented paper scientific discusses".

Point 1: The main problem of its current form is figures quality. Figure 2,3,6 are blurry.

Response 1: We have improved the quality of the Figures 2 and Figure 3 by highlighting the axes, numbers and texts in the images. Figure 6 has been completely redesigned.

Point 2: Additionally, I recommend extensive English check, e.g., in the first sentence of abstract: "The presented paper scientific discusses".

Response 2: We have improved English in two ways: 1. our own thorough reading and 2. a professional service in MDPI "English editing".

All edits are marked in the text of the word file using the "Review / New Comment" menu.